# c-Abl Tyrosine Kinase Is Required for BDNF-Induced Dendritic Branching and Growth

**DOI:** 10.3390/ijms24031944

**Published:** 2023-01-18

**Authors:** América Chandía-Cristi, Nicolás Stuardo, Cristian Trejos, Nancy Leal, Daniela Urrutia, Francisca C. Bronfman, Alejandra Álvarez Rojas

**Affiliations:** 1Cell Signaling Laboratory, Department of Cellular and Molecular Biology, Center for Aging and Regeneration (CARE), Millennium Institute on Immunology and Immunotherapy, Biological Sciences Faculty, Pontificia Universidad Católica de Chile, Portugal 49, Santiago 8330025, Chile; 2Neurosignaling Lab, Center of Aging and Regeneration (CARE), Institute of Biomedical Science (ICB), Faculty of Medicine, Universidad Andrés Bello, Echaurren 183, Santiago 8370071, Chile

**Keywords:** dendritic arborization, c-Abl, BDNF/TrkB pathway

## Abstract

Brain-derived neurotrophic factor (BDNF) induces activation of the TrkB receptor and several downstream pathways (MAPK, PI3K, PLC-γ), leading to neuronal survival, growth, and plasticity. It has been well established that TrkB signaling regulation is required for neurite formation and dendritic arborization, but the specific mechanism is not fully understood. The non-receptor tyrosine kinase c-Abl is a possible candidate regulator of this process, as it has been implicated in tyrosine kinase receptors’ signaling and trafficking, as well as regulation of neuronal morphogenesis. To assess the role of c-Abl in BDNF-induced dendritic arborization, wild-type and c-Abl-KO neurons were stimulated with BDNF, and diverse strategies were employed to probe the function of c-Abl, including the use of pharmacological inhibitors, an allosteric c-Abl activator, and shRNA to downregulates c-Abl expression. Surprisingly, BDNF promoted c-Abl activation and interaction with TrkB receptors. Furthermore, pharmacological c-Abl inhibition and genetic ablation abolished BDNF-induced dendritic arborization and increased the availability of TrkB in the cell membrane. Interestingly, inhibition or genetic ablation of c-Abl had no effect on the classic TrkB downstream pathways. Together, our results suggest that BDNF/TrkB-dependent c-Abl activation is a novel and essential mechanism in TrkB signaling.

## 1. Introduction

Regulation of dendritic arborization is essential for brain function since it determines the receptive field of neurons [1]. Extrinsic neuronal signals that promote dendritic development include soluble ligands such as neurotrophins that bind to specific membrane receptors. Brain-derived neurotrophic factor (BDNF) binds to the tyrosine kinase receptor TrkB [2] promoting neurite extension, arborization and growth of dendrites, the formation and establishment of synaptic connections, survival, and differentiation of neuronal populations [3,4,5,6,7]. The binding of BDNF to TrkB stimulates its dimerization and transphosphorylation at Tyr residues, which in turn recruits signaling adaptors to transduce the signal inside the cell. The classical downstream signaling pathways triggered after TrkB-activation are the mitogen protein-activated protein kinases (MAPK), phosphoinositide-3-kinase (PI3K), and phospholipase-C gamma (PLC-γ) pathways. After TrkB activation by its ligand, the receptor undergoes endocytosis and transport to the neuronal soma. The internalized neurotrophin bound to its Trk receptor continues signaling in endosomes by activating components of the Ras-MAP kinase, PLC-γ, and PI3K pathways, thus participating in long-range signaling and transcriptional events [8]. Subsequently, the signaling endosome follows its post endocytic transport, and the receptor can be recycled into the plasma membrane or degraded by lysosomes [9,10,11,12].

The process of dendritic growth and branching requires active reorganization of the actin cytoskeleton. Neurite outgrowth has been proposed to be directly regulated by lamellipodia and filopodia formation, which in turn are dependent on actin cytoskeleton dynamics mediated by Rho GTPases such as Rac1, Cdc42, and RhoA [13,14]. Interestingly, BDNF-TrkB signaling has been shown to regulate the dynamics of actin filaments by modulation of Rho GTPase family proteins and other actin remodeling proteins [15,16,17]. However, the specific mechanisms that coordinate the dendritic outgrowth induced by BDNF/TrkB signaling have not yet been fully elucidated. A possible mediator of this process is the ubiquitous non-receptor tyrosine kinase c-Abl, a key signal transducer for different kinds of stimuli including cellular insults such as DNA damage and oxidative stress, and for receptors of extracellular cues involved in axon-guidance, cell adhesion, and cell growth [18]. Several characteristics of c-Abl dynamics and function suggest that it could mediate downstream effects of the BDNF/TrkB pathway. First, c-Abl is activated by growth factors [19] and has been shown to interact bilaterally with multiple tyrosine kinase receptors including EGFR, PDGFRB, EPHB2, and a member of the Trk family of receptors, TrkA [20,21]. Indeed, c-Abl acts as an adapter for TrkA signaling [22]. Secondly, c-Abl has G- and F-actin binding domains and has been shown to interact with multiple actin-remodeling proteins in diverse cell types, including neurons [23]. The c-Abl modulates neuronal morphogenesis and is required for neurite outgrowth and axon guidance [24,25,26,27,28,29,30,31,32]. In primary cultures of hippocampal neurons, c-Abl kinase has been shown to participate in the development of dendritic ramifications by remodeling actin microfilaments [31]. Even though both c-Abl and BDNF/TrkB regulate dendritic morphogenesis in central nervous system neurons, there are so far no studies addressing the functional relationship between both signaling pathways.

Here, we show that BDNF-induced dendritic branching requires c-Abl expression and activity. Interestingly, BDNF increases c-Abl activity and c-Abl-TrkB interaction. Although this process requires TrkB activity, it does not require the activation of MAPK, PI3K, and PLC-γ signaling pathways. These results suggest that c-Abl activation is a key downstream mechanism required for BDNF-TrkB-induced dendritic growth, and its mechanism of action does not involve the regulation of the classical signaling pathways regulated by BDNF/TrkB. To obtain further insight into the mechanism of action of c-Abl activity in the context of BDNF/TrkB signaling and function, we investigated the role of c-Abl in the regulation of BDNF/TrkB internalization and endocytic trafficking. We found that c-Abl activity was not required for BDNF-induced endocytosis or degradation of the TrkB receptor. However, we found that c-Abl was required for basal endocytosis of TrkB and that its activation increased the speed of TrkB retrograde traffic.

Altogether, our results show that c-Abl activation downstream of BDNF/TrkB signaling mediates a cellular process leading to dendritic growth that does not involve the activation of classical signaling pathways triggered by BDNF/TrkB but is required in BDNF/TrkB-dependent dendritic growth.

## 2. Results

### 2.1. c-Abl Is Required for BDNF-Induced Dendritic Arborization

BDNF induces activation of TrkB, promoting dendritic growth [5,15,33,34]. In addition, it has been described that the c-Abl kinase participates in dendritogenesis [31]. However, whether there is crosstalk between c-Abl and BDNF/TrkB signaling is not known. To examine the role of c-Abl tyrosine kinase in BDNF/TrkB signaling, we evaluated BDNF-induced dendritic growth in DIV 7 low-density culture hippocampal neurons after 48 h of incubation with BDNF (50 ng/mL) in the presence of two well-described c-Abl-inhibitors with different binding sites (5 µM Imatinib or GNF2) [35].

As shown in previous studies, neurons treated with BDNF showed increased dendritic branching compared to control neurons (Figure 1A). We assessed the complexity of the dendritic arbor using Sholl analysis, a classical tool to study changes in the dendritic arbor [36]. After BDNF treatment, neurons presented a significant increase in dendrite intersections between 10 and 30 µm away from the soma (Figure 1B). Furthermore, we found that the branching points in the BDNF-treated neurons were twofold higher than those in control neurons (Figure 1D). This increase promoted by BDNF is dependent on the TrkB receptor, since when we used K252a, an inhibitor of Trk receptors, the BDNF-induced increase in intersections and the number of branches was prevented (Appendix A). Therefore, BDNF treatment stimulates dendrite growth and arborization as we have previously shown [8,37,38]. Interestingly, when neurons were treated with BDNF in the presence of the c-Abl inhibitors, Imatinib or GNF2, BDNF-induced dendritic branching was reduced to similar levels of unstimulated neurons (Figure 1A). Neurons treated only with c-Abl inhibitors showed a slight decrease in dendritic arbor complexity as compared to the DMSO condition (Figure 1C). Furthermore, Imatinib and GNF2 prevented the increase in dendritic growth and branching induced by BDNF (Figure 1B,D) as well as the increase in the number of primary dendrites (Figure 1E). Therefore, c-Abl inhibition abolished BDNF-promoted enhancement of dendritic growth and complexity.

To confirm the requirement of c-Abl expression in the dendritic arborization increase induced by BDNF, we used hippocampal neurons derived from c-Abl KO embryos. Seven DIV c-Abl KO and WT hippocampal neurons were treated with BDNF for 48 h, and primary neurites and branching points were evaluated. Consistent with the results obtained with c-Abl inhibitors, the absence of c-Abl expression in neurons reduced BDNF-induced dendritic growth to basal levels (Figure 1F). The Sholl analysis as well as the number of branching points quantifications confirmed this observation (Figure 1G,H).

It has been reported that *abl/arg* double-null mice exhibit defective neural tube morphology and gross alterations in neuroepithelial actin cytoskeleton structures, suggesting that c-Abl knockout neurons could have alterations in dendritic tree development [25]. To discard any confounding effects attributable to developmental alterations in the dendritic arbor of c-Abl KO mouse neurons, we knocked down c-Abl expression by transfecting a shRNA targeting this protein into WT neurons, and then we evaluated the neuronal morphology after BDNF treatment. Consistently, hippocampal neurons transfected with the c-Abl-targeting shRNA did not increase their dendritic branching in response to BDNF treatment (Appendix A).

Interestingly, neurons treated with a c-Abl activator (DPH) showed more branched dendrites compared to control neurons (Figure 1J). After 48 h of DPH treatment, neurons showed a significant increase in dendrite intersections between 10 and 75 µm of the soma (Figure 1K). In addition, DPH increased the number of branching points (Figure 1L) and primary dendrites (Figure 1L). Consistent with the reported role of c-Abl activation on apoptosis induction, we observed that some neurons died after 48 h of treatment with DPH. However, we verified that DPH only produced a 10% reduction of viability in MTT assays (Appendix A). Thus, the use of two different pharmacological inhibitors of c-Abl and two different approximations to reduce c-Abl expression demonstrated that c-Abl activity and expression are required for BDNF/TrkB-induced dendritic arborization in vitro and suggests that c-Abl activation is required for BDNF/TrkB-induced dendritic branching.

### 2.2. BDNF Increases c-Abl Activation and the Interaction of TrkB with c-Abl

It has been described that c-Abl interacts with the TrkA receptor in the juxtamembrane region of TrkA [39]. To explore a potential interaction between the TrkB receptor and c-Abl, we performed immunoprecipitation experiments to evaluate whether c-Abl associates with TrkB after BDNF stimulation. Interestingly, BDNF promoted an interaction between the TrkB receptor and c-Abl (Figure 2A). No association was observed when the immunoprecipitation assay was performed with a control IgG. To determine whether c-Abl is activated in response to BDNF, we examined c-Abl phosphorylation at tyrosine residue 412 [40,41] by immunostaining. Immunofluorescence against phosphorylated c-Abl showed that BDNF (50 ng/mL BDNF for 30 min) increased c-Abl phosphorylation in the soma and the dendrites of hippocampal neurons (Figure 2B). In addition, we confirmed the effect of BDNF on c-Abl activation by western blot analysis. Neurons stimulated with BDNF showed a significant increase in the levels of c-Abl phosphorylation at tyrosine 412 after 30 min (Figure 2C,D). Concomitant with c-Abl activation by BDNF, we observed a significant increment in the phosphorylation of one of the best-characterized substrates of c-Abl, the Crk adaptor protein (phospho-CrkII Tyr221) after 15 min of BDNF stimulation (Figure 2E). We used 5 µM DPH, a c-Abl activator, for 60 min as a positive control to induce c-Abl activity. Even though DPH induced dendritic branching (Figure 1J–L), it did not increase TrkB phosphorylation (Figure 2C). These results collectively suggest that BDNF activates c-Abl directly downstream of TrkB activation.

### 2.3. BDNF-Induced c-Abl Activation Requires TrkB Activity but Does Not Involve the TrkB Downstream Signaling Pathways Regulated by PI3K, ERK1/2, and PLC-γ Signaling

Since BDNF increases c-Abl activation and interaction with TrkB, we investigated whether TrkB signaling induces c-Abl activation. To this end, neurons were exposed to BDNF and co-treated or not with the pan-Trk tyrosine kinase inhibitor K252a (Figure 3A), and c-Abl phosphorylation was evaluated by western blotting. The increase in phosphorylation at Tyr 412 of c-Abl induced by BDNF was abolished by the K252a inhibitor, suggesting that Trk kinase activity is required for c-Abl phosphorylation. Furthermore, we evaluated BDNF-induced c-Abl phosphorylation in hippocampal neurons derived from TrkB^F616A^ mouse embryos, which express a modified TrkB receptor whose activity is sensitive to the kinase inhibitor 1NM-PP1. Neuronal cultures were pre-treated for 1 h with 1NM-PP1 (1 μM) and next, incubated with 50 ng/mL BDNF in the presence or absence of 1NM-PP1. In agreement with the results obtained with K252a, we observed that 1NM-PP1-mediated TrkB inhibition prevented the activation of c-Abl induced by BDNF, which was previously observed by immunofluorescence and western blotting analysis (Figure 3B,C). These results confirm that BDNF-induced c-Abl activation is TrkB-dependent.

The phosphorylation sites on the TrkB cytosolic tail serve as specific anchoring sites for different intracellular adapter proteins, which contain Src 2 (SH2) and phosphotyrosine-binding domains (PTB). These adapter proteins mediate the activation of TrkB downstream signaling pathways, including RAS/MAPK (ERK1/2), PI3K/Akt, and PLC-γ/PKC [41]. To investigate whether c-Abl phosphorylation is dependent on one of these pathways, we evaluated the activation of c-Abl after BDNF stimulation in the presence or absence of inhibitors for these TrkB downstream signaling pathways. Surprisingly, BDNF-induced phosphorylation of c-Abl was observed in the presence of the neuronal treatments with LY294002 a PI3K inhibitor (Figure 3D), U73122, an ERK1/2 inhibitor (Figure 3E) or U0126, a PLC-γ inhibitor (Figure 3F). These inhibitors effectively reduced the target signaling pathway since we observed a dose-dependent reduction of Akt, PKC, and ERK1/2, phosphorylation. Therefore, the activation of the ERK1/2, PI3K, and PLC-γ pathways is not involved in the BDNF/TrkB-induced c-Abl activation, suggesting a direct activation of c-Abl by TrkB.

### 2.4. c-Abl Restricts Basal Levels of TrkB and Downregulates Surface TrkB Levels but Does Not Affect BDNF-Induced Endocytosis

To obtain further insight into the mechanism by which c-Abl allows BDNF/TrkB-induced dendritic branching, we examined the requirement of c-Abl activity for TrkB endocytosis and endocytic trafficking. We started by evaluating whether c-Abl activity affects the levels of TrkB receptors in neurons after BDNF stimulation. We evaluated TrkB receptor levels in neurons treated with the c-Abl inhibitor Imatinib (5 µM) and performed activation time curves with BDNF (50 ng/mL). We observed an increased basal level of TrkB when c-Abl activity was reduced. Neurons maintained this trend at 1 h of stimulation with BDNF; however, at longer times no differences were observed in the levels of the TrkB receptors (Figure 4A). Along the same lines, we observed that the basal levels of TrkB in c-Abl KO neurons were increased compared to WT neurons. When we stimulated with 50 ng/mL of BDNF, the decrease in TrkB levels over time did not show differences between neurons with or without c-Abl expression (Figure 4B).

Then, we explored if the increased basal levels of TrkB in c-Abl null neurons or during c-Abl inhibition were due to alterations in TrkB receptor endocytic trafficking. We pre-treated neurons with c-Abl inhibitors (Imatinib or GNF-2) for 1 h and then stimulated with BDNF for 30 min and measured levels of TrkB receptor on the plasma membrane using a surface biotinylation assay. The basal levels of TrkB increased 2 and 1.5 times by the treatments with the c-Abl inhibitors Imatinib and GNF2, respectively, compared to control (Figure 4C,D). Interestingly, BDNF treatment induced a reduction of TrkB receptors in the plasma membrane in control neurons and in the neurons treated with the c-Abl inhibitors, and both conditions reached a similar amount of surface TrkB receptor by the end of the BDNF stimulus. This result shows that TrkB internalization in response to BDNF increases significatively (3-fold) when c-Abl is inhibited.

We then investigated whether TrkB receptors located in the plasma membrane were sorted to the lysosomes in the presence or absence of c-Abl after BDNF treatment using an antibody feeding assay [37]. To evaluate this, we transfected c-Abl-KO neurons with a Flag-TrkB coding plasmid, and after 24 h of expression we treated the transfected neurons with a Flag-specific antibody at 4 °C (20 min). The excess antibody was then washed off, and the neurons were stimulated with BDNF for 3 h at 37 °C. After washing and fixation, we evaluated the co-localization of Flag immunostaining with an antibody against Lamp1. The colocalization of TrkB and Lamp1 signals indicates TrkB presence in lysosomes. Although a slight, non-statistically significant decrease in TrkB and Lamp1 colocalization was observed in c-Abl-KO neurons compared to WT neurons in basal conditions, no significant differences were observed after BDNF treatment (Figure 4E).

Our results show that inhibition of c-Abl or reduction of c-Abl expression increases the levels of TrkB receptor on the cell surface. Consistent with these results, when hippocampal neurons were treated with DPH for 30 min, the c-Abl activator reduced the levels of TrkB in the plasma membrane to the same level as the cells treated with BDNF (Figure 4F). Furthermore, we performed treatments with DPH and BDNF in non-permeabilized neurons to evaluate the amount of endogenous TrkB in the cell surface. This was achieved by using an antibody directed against the extracellular domain of the receptor. As expected, the immunofluorescence quantification showed a decrease in the signal for TrkB (red) in both BDNF-treated neurons (Appendix A) and DPH-treated neurons, suggesting that both treatments induce TrkB internalization. Finally, neurons transfected with TrkB-Flag and labeled with anti-Flag antibodies at 4 °C showed that after 30 min of DPH treatment at 37 °C, TrkB accumulated in intracellular compartments independently of BDNF treatment (Figure 4G). Altogether, these results suggest that in the absence of BDNF, the basal levels of c-Abl activity downregulate the levels of TrkB receptors at the cell surface.

### 2.5. BDNF Downstream Signaling Is Not Affected by c-Abl Activity

Although c-Abl expression and activity regulates the trafficking of TrkB in the absence of BDNF, it did not seem to regulate BDNF-dependent internalization and intracellular trafficking and does not explain the requirement of c-Abl for BDNF-TrkB-dependent enhancement of dendritic arborization. Since c-Abl phosphorylation was not regulated by TrkB downstream signaling pathways, we evaluated whether c-Abl is upstream of ERK1/2, PI3K, or PLC-γ. It is well known that the ERK1/2 and PI3K signaling pathways are required for BDNF-induced dendritic growth after TrkB activation [42,43]. To pursue this aim, we first analyzed if c-Abl activity modulates BDNF-induced TrkB activation. The increase of TrkB phosphorylation at Tyrosine 515 induced by BDNF was observed both in control neurons and in the presence of the c-Abl inhibitors, Imatinib or GNF2 (Figure 5A,B). This suggests that c-Abl activity is not required for TrkB activation. Next, we focused on PI3K/Akt and ERK1/2 phosphorylation induced by BDNF. Interestingly, the presence of Imatinib and GNF2 did not affect the activation of these signaling pathways (Figure 5C,D).

To confirm these results with a loss of function approach we evaluated the phosphorylation levels of TrkB, ERK1/2, and Akt in response to BDNF treatment in c-Abl KO neurons. Consistently, with our previous results using c-Abl inhibitors, hippocampal c-Abl-KO neurons showed similar levels of activation of TrkB in the presence of BDNF compared to wild-type neurons (Figure 5E,F). Furthermore, the magnitude and time course of TrkB, Akt, and ERK1/2 activation induced by BDNF (0, 1, 3, or 6 h) were similar in WT and c-Abl null neurons (Figure 5G,H). Therefore, c-Abl function is not required for the activation of classical BDNF-TrkB downstream signaling cascades.

### 2.6. c-Abl Increases the Retrograde Movement of TrkB Vesicle in Neurites

Studies have established that the post-endocytic trafficking of TrkB vesicles is required for dendritic arborization, by increasing local signaling in dendrites and synapses [37,38,44,45,46]. To study the effect of c-Abl on TrkB post-endocytic vesicular trafficking, hippocampal neurons were co-transfected with a c-Abl-GFP construct to overexpress c-Abl and TrkB-mCherry. Then, live cell imaging was performed to assess the anterograde and retrograde speed of TrkB-mCherry moving vesicles. The quantification of the vesicular transport speeds showed that c-Abl overexpression increased the speed of TrkB vesicles moving in the retrograde direction but had no effect on anterograde speed (Figure 6A). To confirm that c-Abl affects TrkB vesicular velocity, c-Abl-KO neurons were transfected with TrkB-mCherry fusion protein, and the same analysis of live cell images was performed. Interestingly, the retrograde speed of TrkB-mCherry fusion protein was significantly reduced in c-Abl knockout neurons as compared to wild-type neurons (Figure 6B). In addition, when analyzing the TrkB-m-Cherry vesicles transfected in c-Abl-KO neurons, we observed TrkB vesicles with elongated and tubular shapes, different from the circular morphology of the vesicles in WT neurons. (Figure 6C). Taken together, analysis of TrkB vesicle trafficking suggests that c-Abl activity is required for an efficient retrograde TrkB transport rate in neurites.

## 3. Discussion

Development and maintenance of the dendritic arbor are crucial for proper brain function [47]. It is known that the c-Abl tyrosine kinase participates in dendrogenesis [30,31]; however, the signaling pathways that regulate its activation are not fully understood. In the present study, we identified c-Abl as a new TrkB receptor downstream signal component required for the inductive effect of BDNF on dendritic growth and arborization. Consistent with the literature, the c-Abl activator DPH induced an increase in the arborization and growth of dendritic trees. Interestingly, we observed that two c-Abl inhibitors that have different inhibition mechanisms, Imatinib and GNF2, have the same effect on preventing BDNF-induced dendritic arborization. Furthermore, we observed similar results using neurons transfected with a shRNA targeting c-Abl, and c-Abl-KO neurons, confirming the need for c-Abl activity for BDNF-induced TrkB activation to effectively promote dendritic arborization. Using western blot and immunofluorescence in TrkBF616A hippocampal neurons, we observed that BDNF promotes c-Abl phosphorylation in a TrkB-dependent manner. Interestingly, ERK1/2, PI3K, and PLC-γ signaling pathways were not required for c-Abl activation, suggesting that c-Abl acts through an independent pathway downstream of TrkB. Moreover, by surface membrane biotinylation, we observed that inhibition of c-Abl activity increases the availability of TrkB in the membrane in the absence of its ligand. This process did not affect BDNF-dependent endocytosis of the receptor or BDNF-induced sorting to lysosomes. Together, our results suggest that BDNF/TrkB-dependent c-Abl activation is a novel mechanism that is essential for effective TrkB signaling leading to dendrite arborization. Moreover, our results indicate that c-Abl function downstream of BDNF/TrkB is independent of the more studied TrkB downstream pathways: MAPK, PI3K, and PLC-γ. It is possible that other c-Abl-regulated processes are contributing to BDNF-dependent dendritic branching including regulation of cytoskeleton and postendocytic trafficking of the receptor. Consistent with this last possibility, we found that the absence of c-Abl decreases the retrograde vesicular speed of the TrkB receptor, and, oppositely, the overexpression of an active c-Abl–GFP increases the retrograde vesicular speed of the receptor. Retrograde vesicular TrkB receptor transport has been described as a key event for BDNF-TrkB-induced dendritic arborization.

Reciprocal interactions between c-Abl and tyrosine kinase receptors, including EGFR, PDGFRB, and EPHB2, have been demonstrated in previous studies [20,21,22]. Interestingly, studies in HEK293 cells overexpressing c-Abl and TrkA have shown that these two proteins interact in the juxtamembrane region of the TrkA receptor [39]. This interaction has also been demonstrated in yeast in two hybrid experiments [48]. This suggests that the interaction between c-Abl and members of the Trk receptors family is a conserved mechanism among these two kinase proteins. Since we only observed TrkB-c-Abl association under BDNF stimulation, we could speculate that c-Abl is a signal transducer for BDNF-TrkB signaling. Consistently, using a pan-Trk inhibitor or neurons derived from TrkBF616A knock-in mice, we showed that BDNF promotes c-Abl activation and CrkII phosphorylation, a downstream target of c-Abl. Similarly, studies in PC12 cells have shown that NGF induces CrkII phosphorylation in a c-Abl-dependent manner, impacting processes of cellular morphogenesis [23]. However, Crk is not required for BDNF-induced dendritic arborization [49], suggesting that TrkB-c-Abl signals through other pathways to regulate dendritic arborization. To elucidate the pathway involved in c-Abl activation, we pharmacologically inhibited each classic downstream pathway of TrkB including MAPK, AKT, and PLC-γ. To our surprise, the inhibition of these pathways did not prevent c-Abl phosphorylation induced by BDNF, and vice versa, c-Abl inhibition did not affect the activation of the TrkB signaling pathways. Altogether, our findings suggest that c-Abl plays a role in the regulation of neuronal morphology as an independent downstream pathway triggered by BDNF activation of TrkB. Further experiments are required to elucidate the complete TrkB–c-Abl signaling axis and the precise molecular mechanisms it activates to mediate dendritic growth.

One fundamental question in the neurotrophic factor field is how neurotrophins coordinate all the complex cellular processes that are involved in neurite growth, such as receptor endocytosis, endosomal trafficking, and cytoskeleton dynamics regulation. c-Abl has been implicated in all three of these processes, making it a very interesting candidate for downstream mediation of TrkB activation.

Endocytosis is a highly dynamic processes that requires tight regulation of the cytoskeleton. The c-Abl tyrosine kinase has been implicated in receptor tyrosine kinase signaling and endocytosis in non-neuronal cell types. For example, c-Abl stabilizes the EGF receptor (EGFR) in the membrane, negatively affecting its endocytosis and traffic to the lysosomes in a ligand-dependent manner [20]. c-Abl has also been shown to stabilize the transferrin receptor in the cell membrane of MEF cells [50]. On the other hand, c-Abl positively regulates BCR internalization through CrkII phosphorylation and Rac1 activation [51], demonstrating that c-Abl is capable of regulating growth factor receptor endocytosis via the regulation of actin dynamics. When we biotinylated surface membrane proteins in neuronal cultures, both Imatinib and GNF2 considerably increased the available levels of TrkB in the membrane. However, the inhibition of c-Abl did not affect BDNF-promoted receptor internalization, suggesting that c-Abl restricts the availability of TrkB in the plasma membrane independently of BDNF. Consistently with the increase of receptor availability, when c-Abl was inhibited, we observed an over-activation of the canonical TrkB downstream signaling pathways. An enticing possibility is that c-Abl could be regulating TrkB endocytosis via activation of cyclin-dependent kinase 5 (Cdk5). We and others have shown c-Abl can activate Cdk5 [27] and mediate physiological and pathophysiological processes [52], but their joint contribution to BDNF-induced dendritic branching has not been evaluated to date. Our results show that BDNF induces an increase in active Cdk5 (phosphorylated on Tyr15) which can be partially prevented by inhibiting c-Abl with GNF-2, an allosteric inhibitor (Appendix A). Cdk5 plays key roles in tyrosine kinase receptor endocytosis by regulating endophilin-associated machinery [53]. BDNF-induced TrkB endocytosis is regulated through a tripartite interaction between Endophilin A1, retrolinkin, and the WAVE1 complex [54], suggesting a possible role for Cdk5 in specifically regulating TrkB endocytosis. Normal Trkb signaling can be disrupted by prolonged phosphorylation or lack of internalization of the receptor [10,55,56], suggesting that the increase of membrane-associated TrkB availability could be detrimental for proper BDNF signaling. The deregulated activation of c-Abl in disease states could sequester TrkB in the cell by inhibiting BDNF/TrkB signaling.

Post-endocytic trafficking of Trk receptors is required for proper signaling and neuronal function [11,57]. Indeed, several lines of evidence have shown that the effects of BDNF/TrkB on the activation of key downstream targets are dependent on sustained signaling from endosomes after internalization of BDNF/TrkB [8,58,59]. For example, it has been shown that Rab5 and Rab11 activity is required for BDNF-mediated dendritic branching as it maintains persistent activation of TrkB downstream signaling pathways [8,38]. Dendritic Rab5-positive early endosomes co-localize with TrkB and increase retrograde movement in response to BDNF treatment, most likely to convey the trophic signal to the nucleus and promote the transcription of neuronal growth and plasticity-related genes. Rab11 was also shown to be necessary to maintain TrkB phosphorylation, ERK1/2 activation, and CREB phosphorylation, all of which are required for BDNF to promote dendritic arborization. Interestingly, the expression of a Rab11 dominant-negative mutant significantly changes the transcriptional response to BDNF, decreasing the mRNA levels of proteins required for dendritic development, such as Arc [60]. Therefore, a possible explanation for the changes observed in dendritic arborization when manipulating c-Abl activity and expression could be related to changes in post-endocytic trafficking dynamics. When we evaluated total TrkB levels in the absence of c-Abl activity, we observed an upregulation of the receptor’s levels, which could be explained by decreased endosomal trafficking into lysosomes. This observation prompted us to evaluate post-endocytic trafficking dynamics of TrkB in c-Abl neurons. Interestingly, we found that c-Abl-null neurons present a decreased retrograde speed in dendrites and a tendency towards a decrease in the location of TrkB in Lamp1 positive endosomes, and conversely, c-Abl overexpression increases retrograde trafficking in dendrites. Therefore, active c-Abl appears to promote retrograde trafficking of TrkB, which is a critical step in the process of BDNF-promoted dendritic arborization. A possible explanation for the increase in retrograde trafficking could be c-Abl-mediated regulation of molecular motors. While there are no reports of molecular motor–c-Abl interactions in neurons, the downstream target of c-Abl, Cdk5, has been shown to regulate both kinesin [61,62,63] and dynein activity in neurons [64,65]. Given the mixed microtubule polarity present in dendrites [66], a fine balance between the activation of both molecular motors is required for effective retrograde transport to take place. A hypothetical TrkB–c-Abl–Cdk5 signaling axis could regulate post-endosomal trafficking of the receptor to ensure effective BDNF-TrkB signaling.

It has been shown that BDNF induces dendrite branching and filopodial formation in developing neurons through the activation of actin binding proteins to promote actin polymerization [67,68] and Rho GTPase activity [69,70,71]. In this work, we showed that BDNF activates c-Abl to promote dendritic growth. In that context, it has been shown that c-Abl induces neurite growth modulating the dynamics of the actin cytoskeleton, either directly through its domain of interaction with F-actin [24] or indirectly through the inactivation of actin cytoskeletons regulatory proteins such as RhoA [29,31,72,73], Dok1 [74], WAVE [41,75], and Cdk5 [27]. Interestingly, Cdk5 can phosphorylate TrkB directly in response to BDNF, and the effects of this phosphorylation are relayed to the actin cytoskeleton through the attenuation of Cdc42 activity [16] and activation of Rac1 [76]. The activation of this signaling pathway is required for dendritic growth. It is entirely possible that c-Abl mediates actin cytoskeleton destabilization through RhoA inactivation while simultaneously regulating Rac1 and Cdc42 activity via Cdk5 phosphorylation. c-Abl could therefore be mediating tight temporal and spatial control of actin cytoskeleton dynamics to promote growth of the dendritic arbor.

## 4. Materials and Methods

### 4.1. Ethics Statement

All procedures were reviewed and approved by the Bioethics and Care of Laboratory Animals Committee of the Pontificia Universidad Católica de Chile (Protocol #150721002), which follows the local guidance documents generated by the National Research and Development Agency (ANID) and the Guide for the Care and Use of Laboratory Animals published by NIH of US Public Health Service.

### 4.2. Animals

Primary hippocampal and cortical neurons were prepared from rat or mouse embryos obtained from the institutional (CIBEM-UC) vivarium. TrkB^F616A^ knock-in mutant mice were obtained from The Jackson Laboratory and bred in-house as homozygous mating pairs. This strain was generated as previously described [77]. The mice have C57BL/6 genetic background and did not display any gross physical or behavioral abnormalities. Homozygous c-Abl^loxp^/c-Abl^loxp^ mice were kindly donated by Dr. Anthony J Koleske (Yale School of Medicine, US) and bred within our animal facility. c-Abl^loxp^/c-Abl^loxp^ were bred with Nestin-Cre^+^ mice, which were obtained from The Jackson Laboratory. This strain was originated and maintained on a mixed B6.129S4, C57BL/6 background and did not display any gross physical or behavioral abnormalities. The c-Abl null embryos (c-Abl^loxp^/c-Abl^loxp^, Nestin-Cre ^+^, called from here on c-Abl-KO), their siblings c-Abl^loxp^/c-Abl^loxp^ that express c-Abl (called from here on c-Abl-WT), and the TrkB^F616A^ mice were housed at a 12/12 h light/dark cycle at 24 °C with ad libitum access to food and water. Genotyping was performed using PCR-based screening [42].

Primers:Abl1-flox forward: 5′-CCT GGC CTC CAA GAG CAC-3′Abl1-flox reverse: 5′-AGC CCC AGG GCA TAG ATA GT-3′Nes-Cre forward: 5′-TTG CTA AAG CGC TAC ATA GGA-3′ (wild type form)Nes-Cre reverse: 5′-GCC TTA TTG TTG AAG GAC TG-3′Nes-Cre forward: 5′-CCT TCC TGA AGC AGT AGA GCA-3′ (mutant form)Nes-Cre reverse: 5′-GCC TTA TTG TTG AAG GAC TG-3′Ntrk2 forward: 5′-AGG GCA AAA GGG TTG CTC-3′ andNtrk2 reverse: 5′-CCA GCA GAA CAC TCG ACT CA-3′

### 4.3. Antibodies

Commercial antibodies included: anti-TrkB (BD Biosciences, New Jersey, NJ, USA; #cat 610102), anti-TrkB (Millipore, Burlington, MA, USA; #cat 07-225), anti-phospho-TrkB (Tyr 515) (Sigma, St Louis, MO, USA; #cat SAB4503785), c-Abl (Sigma #cat A5844), anti-c-Abl (Santa Cruz Biotechnology, Dallas, TX, USA; #cat K-12), anti-c-Abl (Santa Cruz Biotechnology, #cat 24-11), anti-phospho-c-Abl (Tyr 412) (Sigma, #cat C5240), anti-PLC-γ (Cell Signaling, Danvers, MA, USA; #cat 2822), anti-phospho-PLC-γ (Tyr 783) (Cell Signaling, #cat 2821), anti-AKT (Cell Signaling, #cat 9272), anti-phospho-AKT (Ser 473) (Cell Signaling, #cat 9291), anti-ERK (Cell Signaling, #cat 9102), anti-phospho-MAPK (Erk1/2; p42/44; Thr202/Tyr204) (Cell Signaling, #cat 9101), anti-Flag (Sigma, #cat F3040), anti-GAPDH (Santa Cruz, #cat 6C5), β-III Tubulin (Sigma, #cat T8578), anti-MAP2 (Sigma, #cat M3696), anti-phospho-CrkII (Tyr221) (Cell Signaling, #cat 3491), anti-CrkII (Cell Signaling, #cat 3492), Secondary Antibody: Goat anti-Mouse IgG (H + L) and Goat anti-Rabbit IgG (H + L), HRP (Invitrogen, Carlsbad, CA, USA, #cat 31430 and #cat 31460) and secondary antibody Alexa-555, Alexa-488, Alexa-633 and Alexa 547 (1:1000) (Invitrogen, Carlsbad, CA, USA).

### 4.4. Primary Cultures of Hippocampal and Cortical Neurons

Pregnant rats or mice were euthanized under deep anesthesia according to the bioethical protocols of our institution. Hippocampi or cortex from embryos were dissected in Hank’s balanced salt solution at 4 °C (NaCl 135 mM, KCl 5.4 mM, NaH_2_PO_4_ 0.5 mM, Na_2_HPO_4_ 0.33 mM and D-glucose 5.5 mM), and primary cultures were prepared as previously described [78]. After disaggregation, cells were seeded on poly-L-lysine-coated wells (0.25 mg/mL). To assess neuronal morphology, we performed immunofluorescence of neurons seeded at low density (density of 15 × 10^3^ cells per well in 12 mm coverslips). For biochemical analyses the neurons were seeded (20,000 cells/cm^2^) and maintained for 2 h with DMEM/HS media (Dulbecco Minimum Essential Medium supplemented with 10% horse serum, 1× glutamine and 1× antibiotic). The culture media was then changed to neurobasal medium (Invitrogen, Carlsbad, CA, USA) and supplemented with B27 (Gibco Invitrogen Corporation, Waltham, MA USA), 2 mM L-glutamine, 100 U/mL penicillin, and 100 μg/mL streptomycin (Invitrogen, Carlsbad, CA, USA), and cultures were maintained at 37 °C in 5% CO_2_. On the next day, cultured neurons were treated with 0.25 µg/mL cytosine 1-β-d-arabinofuranoside (AraC, Sigma-Aldrich, St. Louis, MO, USA) to inhibit the growth of glial cells.

### 4.5. Neuronal Treatments

To activate TrkB signaling, 7 DIV hippocampal neurons were deprived of B27 for one hour and then treated with 50 ng/mL BDNF for 5, 15, 30, and 60 min. In addition, DPH, an allosteric c-Abl activator, was used at a final concentration of 5 µM (Sigma Aldrich). To inhibit the TrkB receptor, hippocampal neurons were deprived of B27 and pretreated for 1 h with K252a (Tocris, Bristol, UK) at 0.2 µM, and then stimulated for 30 min with BDNF 50 ng/mL. In addition, 7 DIV hippocampal neurons from TrkB^F616A^ knock-in embryos were used. TrkB^F616A^ knock-in mice have a point mutation introduced into TrkB to convert phenylalanine to alanine at position 616 (F616A), which allows pharmacological and temporal inhibition of TrkB signaling via the highly membrane-permeable small molecule 1NM-PP1 [77]. The 1NM-PP1 inhibitor (0.5 µM, Cayman, Ann Arbor, MI, USA) was applied for 1h in neurobasal media without B27, and then neurons were stimulated for 30 min with BDNF (50 ng/mL). BDNF was maintained throughout all the treatments.

Dendritic arborization was induced in 7 DIV neurons by stimulation with 50 ng/mL of BDNF (Alomone, Jerusalem, Israel) for 48 h. Inhibitors for c-Abl, Imatinib, and GNF2 were used at a final concentration of 5 µM (Novartis, Basel, Switzerland) and for Trk receptors, K252a 0.2 µM (Tocris) was applied 1 h before applying BDNF and maintained throughout all the treatment. The c-Abl activator DPH was applied at a final concentration of 5 µM for 48 h.

For loss of function experiments, we used c-Abl null neurons or reduced c-Abl expression by transfecting neurons with short hairpin RNA plasmids targeting c-Abl (sh-c-Abl) or a control sh-scramble (c-Abl, sc-270357-SH, Santa Cruz Biotechnology, Santa Cruz, CA, USA). For transient transfections, hippocampal neurons were seeded at a density of 30 × 10^3^ cells per well/in 12 mm coverslips and at 5 DIV neurons were transfected for 2 h in serum-free Opti-MEM using the Lipofectamine 2000 reagent (Life Technologies, Inc., Delhi, India) and treated at 7 DIV with BDNF (50 ng/mL).

To assess downstream TrkB signaling, hippocampal or cortical neurons were deprived of B27 and pretreated (or treated with a vehicle) with the following inhibitors at final concentrations of 1.5 or 10 µM: (a) a potent general PI3K inhibitor, (LY294002, Calbiochem), (b) a highly selective inhibitor of both MEK1 and MEK2 (U0126), and (c) a potent PLC-γ inhibitor (U73122). Then, neurons were stimulated with BDNF (50 ng/mL) for 30 min in the presence of the above-mentioned inhibitors.

### 4.6. Immunofluorescence

After treatment, neurons were rinsed twice with ice-cold PBS and fixed with 4% paraformaldehyde containing 4% sucrose in PBS for 20 min at room temperature. Later, cells were permeabilized for 10 min with 0.2% Triton X-100 in PBS and incubated in 3% bovine serum albumin in PBS (blocking solution) for 30 min at room temperature. Immunostaining was performed using primary specific antibodies by overnight incubation at 4 °C. After being washed 3 times in PBS, neurons were incubated with secondary antibodies (1:1000) for 1 h at room temperature. The cell coverslips were mounted in Dako mounting medium (CS703, Agilent Technologies, Santa Clara, CA, USA), visualized using a Nikon Eclipse Ti2 confocal microscope, and processed and quantified using ImageJ software (National Institutes of Health, Bethesda, MD, USA).

### 4.7. Dendritic Arborization Analysis

After treatments and staining of neurons using an anti-Map2 antibody (1:500) and secondary antibody Alexa-555 (1:1000), dendritic arborization was analyzed by Sholl analysis as previously described [36]. The visualization was performed by confocal microscopy using a Nikon Eclipse C2 confocal microscope connected to a computer with NIS-Elements C software. Images were acquired using a 60× objective at 1024 × 1024 pixel resolution, and 5–7 optical slices were captured along the *z*-axis every 0.5 µm. Z-stacks were integrated, and the images were segmented to obtain binary images. Ten concentric circles with increasing diameters (5 µm each step) were traced around the cell body, and the number of intersections between dendrites and circles was counted and plotted for each diameter. Analysis was performed using the ImageJ plugin developed by the Anirvan Gosh Laboratory (http://biology.ucsd.edu/labs/ghosh/software (accessed on 3 March 2014)). The number of total primary dendrites and branching points of all dendrites were manually counted from the segmented images.

### 4.8. Immunoprecipitation

Hippocampal neurons were plated at a density of 10^6^ cells/cm^2^. At 7DIV neurons were incubated in B27-free neurobasal medium and then stimulated with BDNF 50 ng/mL (Alomone) for 1 h. Next, neurons were washed and lysed in immunoprecipitation assay buffer (20 mM Tris, 150 mM NaCl, 10% glycerol, 1% Igepal, 2 mM EDTA) supplemented with inhibitors (1 mM PMSF, 1 mg/mL aprotinin, 10 mg/mL leupeptin, 1 mM Na_3_VO_4_, and 50 mM NaF). Cell lysates were centrifuged at 14,000 rpm for 15 min at 4 °C. Protein quantification was performed using the Pierce^®^ BCA Protein Assay Kit (Thermo Scientific, Waltham, MA, USA). For immunoprecipitation assays, 300–500 µg of total lysates were incubated with 2 µg of anti-c-Abl (K12 Santa Cruz Biotechnology, Santa Cruz, CA, USA) or anti-TrkB (Millipore) antibodies overnight at 4 °C. Complexes were isolated using protein G-Plus agarose (Santa Cruz cat# sc-2002). Immunocomplexes were subjected to SDS-PAGE and transferred to nitrocellulose membranes (Fisher Thermo Scientific) and analyzed by western blot using anti-TrkB (1:1000), anti-c-Abl (1:1000), and anti-GAPDH (1:5000) antibodies.

### 4.9. Western Blotting

Seven DIV neurons, plated at a density of 10^6^ cells/cm^2^, were incubated in B27-free neurobasal medium for 1 h and then treated with 50 ng/mL of BDNF (Alomone). After washing, neurons were lysed in RIPA buffer (50 mM Tris, 150 mM NaCl, 1 mM EGTA, 1 mM EDTA, 0.5% sodium deoxycholate, 1% NP-40, and 0.1% SDS) and supplemented with protease inhibitors cocktail (Roche) and phosphatase inhibitors cocktail (Roche, Basel, Switzerland). The homogenates were cleared by centrifugation at 14.000 rpm for 10 min. Protein extracts (50–70 µg) were loaded onto 10% SDS-PAGE and transferred to nitrocellulose membranes (Fisher Thermo Scientific). The membranes were blocked with 3% BSA in PBS for 1 h at room temperature, followed by overnight incubation at 4 °C with primary antibodies. After incubation with the appropriate HRP-conjugated secondary antibody (1:3000, Thermo Scientific), membranes were incubated with ECL substrate for detection of HRP enzyme activity (Thermo Fisher Scientific) and visualized in a Syngene gel documentation system. Images were quantified by ImageJ analysis (National Institutes of Health). 2–4 biological replicates were performed for every experiment. The number of repetitions for every experiment is stated in the corresponding figure description. The Student’s *t*-test was used for statistical significance assessment.

### 4.10. Live-Cell Imaging of mCherry-TrkB

Six DIV c-Abl-KO and c-Abl-WT mouse neurons were transfected with 0.8 µg of mCherry-TrkB plasmid (the cDNA encoding mCherry-TrkB was kindly provided by Dr. Chengbiao Wu of the University of California, San Diego [UCSD]) for 2 h using Lipofectamine 2000 in B27-free Opti-MEM medium. During imaging, cultures were kept in Tyrode media (124 mM NaCl, 5 mM KCl, 2 mM CaCl_2_, 1 mM MgCl_2_, 30 mM d-glucose and 25 mM HEPES, pH 7.4) at 37 °C. Live-cell imaging was performed on a Nikon Eclipse C2 confocal microscope equipped with a live-cell temperature controller (LCI cu-501) and digital camera connected to a computer NIS-Elements C software. Images of a single neuron transfected with mCherry-TrkB were acquired using a 60× objective at intervals of 7 s for 3 min to establish the basal level of distribution and dynamics. TrkB-m-Cherry signal mobility was examined using the Manual Tracking ImageJ plug-in (National Institutes of Health, Bethesda, MD, USA).

### 4.11. Immunoendocytosis of Flag-TrkB

Hippocampal neurons (5 DIV) were transfected with 0.5 µg of Flag–TrkB plasmid (gift from Prof. Francis Lee, NYU, USA). After 48 h, hippocampal neurons were incubated at 4 °C for 10 min and then treated with an anti-Flag mouse antibody (1:750) for 20 min. Neurons were washed briefly with warm neurobasal medium and incubated with BDNF (50 ng/mL) for 30 min. Then, neurons were fixed with 4% paraformaldehyde containing 4% sucrose in PBS at room temperature and incubated with a donkey anti-mouse IgG conjugated to Alexa555 (1:1000) without permeabilization. Finally, samples were blocked and permeabilized as described above and immunostained with a donkey anti-mouse IgG-Alexa488 (1:500).

### 4.12. Surface Biotinylation Assay

Hippocampal neurons were pre-treated with GNF2 or Imatinib 5 µM for 1h and then treated with BDNF (50 ng/mL) for 30 min. Neurons were then transferred to 4 °C, rinsed with chilled PBS, and incubated with 0.8 mg/mL biotin (EZ-link^®^ Sulfo-NHS-LC Biotin, Thermo Scientific, Waltham, MA, USA) for 30 min. Subsequently, biotin was quenched with 50 mM NH_4_Cl for 10 min, cells were rinsed two times with PBS, lysed in RIPA buffer (150 mM NaCl, 1% Nonidet p-40, 0.5% sodium deoxycholate, 0.1% SDS, 50 mM Tris, pH 8) and protease (Complete Mini Protease Inhibitor Cocktail; Roche) and phosphatase inhibitors (PhosSTOP Phosphatase Inhibitor Cocktail; Roche), and centrifuged for 1 min, 14,000 rpm at 4 °C. Each lysate was incubated overnight with 30 µL of NeutrAvidin-coupled agarose beads (NeutrAvidin Agarose Resins; Thermo Scientific). Beads were washed with ice-cold lysis buffer, and then biotinylated proteins were eluted with 2 × SDS sample buffer. Cell-surface and total protein lysates were subjected to SDS-PAGE and western blot analysis.

### 4.13. MTT Assay

Cortical neurons were seeded 40.000 cells/well in a 96-well plate and maintained in neurobasal medium (Gibco Invitrogen Corporation) supplemented with B27 (Gibco Invitrogen Corporation) plus 1% penicillin–streptomycin (Gibco Invitrogen Corporation) for 7 days. Then, the cells were treated with 5 µM DPH for 24 h. Cell viability was measured by the modified 3-[4,5-dimethylthiazol-2-yl]-2,5-diphenyltetrazolium bromide (MTT) assay. Then, the MTT reagent (5 mg/mL) was added and incubated for 4 hrs. The medium was then removed and replaced with lysis buffer (50% Dimethylformamide and 20% SDS) overnight. Subsequently, the absorbance of the sample was measured at 620 and 570 nm.

### 4.14. Statistical Analyses

All data are presented as the mean ± Standard Error of the Mean (SEM). Sholl’s analysis curves were compared with two-way repeated measures ANOVA, followed by Bonferroni’s multiple comparisons. Statistical analyses were performed using a Student’s *t*-test or one-way ANOVA followed by appropriate multiple comparisons test depending on the number of groups used in each experiment. Details of the specific tests used, level of significance, and number of replicates are indicated in each figure legend. Statistical analyses were performed using GraphPad Prism 8 (Scientific Software). The significance level was *p* < 0.05 for all treatments.

## Figures and Tables

**Figure 1 ijms-24-01944-f001:**
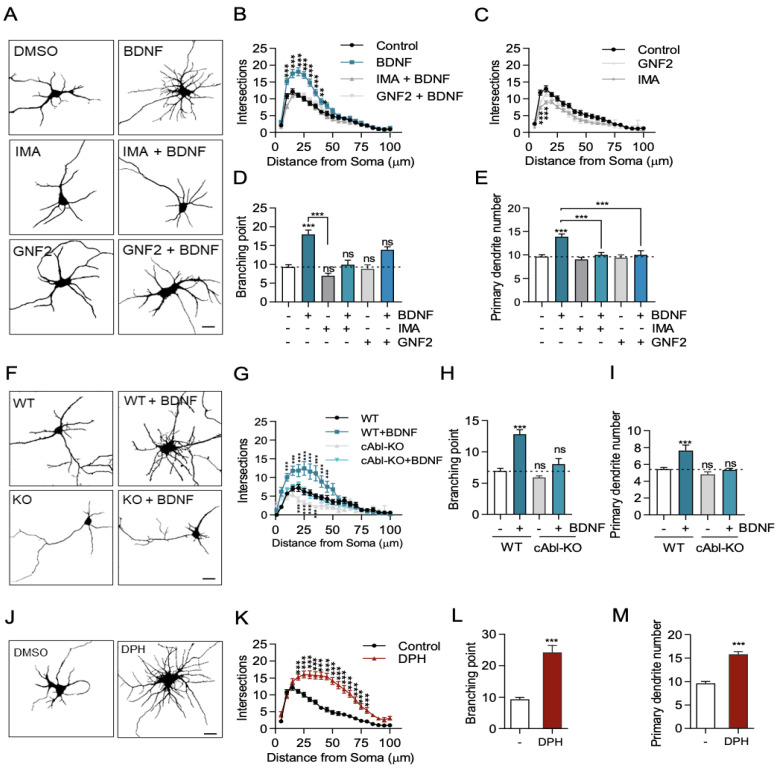
c-Abl is required for BDNF-induced dendritic growth. (**A**–**F**) 7 DIV rat primary hippocampal neurons were stimulated for 48 h with 50 ng/mL BDNF. The neurons were fixed, and MAP2 immunostaining was performed to visualize the somatodendritic compartment. Images were captured on a Zeiss Axiovert 2000 inverted confocal microscope. (**A**) Representative images of hippocampal neurons under conditions of DMSO (control), BDNF, c-Abl inhibitors or BDNF + c-Abl inhibitors. Scale bar = 10 µm. (**B**) Sholl analysis of neurons treated with BDNF or BDNF + c-Abl inhibitors. (**C**) Sholl analysis of neurons treated with c-Abl inhibitors. (**D**) Branching point analysis quantification: control (9.3 ± 0.6), BDNF (18 ± 1.1); IMA (7.0 ± 0.7); IMA + BDNF (9.9 ± 1.2); GNF2 (8.8 ± 1.1); GNF2 + BDNF (13.9 ± 0.6). (**E**) Primary dendrite outgrowth quantification in neurons in the different experimental conditions described in A: Control (9.6 ± 0.4), BDNF (13.9 ± 0.6), IMA (9.0 ± 0.4), IMA + BDNF (10.0 ± 0.5), GNF2 (9.4 ± 0.6), and GNF2 + BDNF (10.0 ± 0.9). (**F**) c-Abl KO neurons do not respond to BDNF-induced dendritic branching. Representative images of c-Abl-KO and WT mouse primary hippocampal neurons treated with 50 ng/mL BDNF for 48 h. Scale bar = 10 µm (**G**) Sholl analysis of control and BDNF-treated WT and c-Abl KO neurons. (**H**) Branching point analysis quantification: WT (6.9 ± 0.4), WT + BDNF (12.8 ± 0.7); c-Abl-KO (5.9 ± 0.6); c-Abl-KO (8.0 ± 0.9). (**I**) Primary dendrite outgrowth quantification in neurons in the different experimental conditions described in F: WT (5.4 ± 0.2), WT + BDNF (7.6 ± 0,7); c-Abl-KO (4.8 ± 0.3); c-Abl-KO (5.3 ± 0.2). (**J**) Representative image of 7DIV rat primary hippocampal neuron treated for 48 h with the c-Abl activator DPH or DMSO (control). Scale bar = 10 µm. (**K**) Sholl analysis of neurons treated with DMSO or DPH. (**L**) Branching point analysis of control and DPH neurons. (**M**) Primary dendrite outgrowth quantification of control and DPH-treated neurons. 40–60 neurons were analyzed per experimental group from three independent experiments. * *p* < 0.05, ** *p* < 0.01, *** *p* < 0.001. Statistical analysis corresponds to a one-way ANOVA test followed by Bonferroni post-test for multiple comparisons and two-way ANOVA with Bonferroni post-test for multiple comparisons. Results are expressed as ±SEM. ns = Not statistically significant.

**Figure 2 ijms-24-01944-f002:**
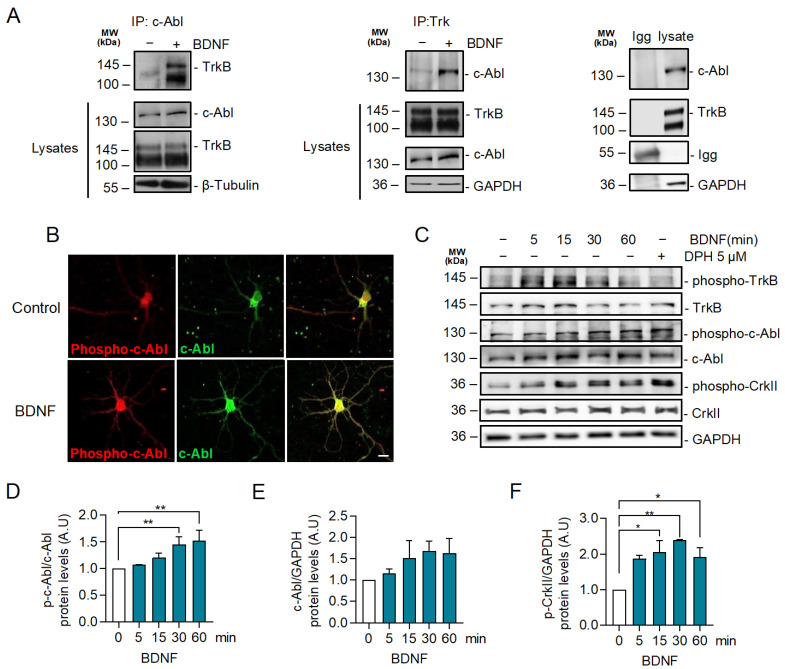
BDNF promotes TrkB-c-Abl interaction and c-Abl activation. (**A**) Seven DIV rat hippocampal neurons were stimulated with 50 ng/mL BDNF for one hour. Cell lysates were prepared and then immunoprecipitation was performed and analyzed by western blotting using incubated anti-c-Abl or anti-Trk antibodies. Immunoprecipitation with a control IgG served as a control for the specificity of the interaction. BDNF-induced coimmunoprecipitation of TrkB and c-Abl. *n* = 2. (**B**) Representative images of immunostaining for Phospho-Tyr214 c-Abl (p-c-Abl) (red) and c-Abl (green) in control and neurons stimulated with BDNF for 30 min. Scale bar = 10 µm. BDNF increased the phosphorylation of c-Abl in the soma and dendrites. (**C**) Neurons were stimulated for different amounts of time (0, 5, 15, 30, and 60 min) with 50 ng/mL of BDNF, and 5 µM of DPH for 60 min was used as a positive control of c-Abl activation. Phospho-c-Abl, c-Abl, Phospho-CrkII (Tyr221), Phospho-TrkB (Tyr515), TrkB, and GAPDH were evaluated by western blot. Stimulation with BDNF resulted in increased levels of p-c-Abl and Phospho-Tyr221 CrkII (p-CrkII). (**D**) Densitometric quantification of p-c-Abl levels normalized against c-Abl levels. The data corresponds to 3 independent experiments. The results are expressed as the mean ± SEM. Statistical analysis corresponds to ANOVA and Tukey post-test (0 min/30 min ** *p* = 0.0044). (**E**) Densitometric quantification of the c-Abl levels normalized to the GAPDH levels. The data corresponds to 3 independent experiments. The results are expressed as the mean ± SEM. Statistical analysis corresponds to ANOVA and Tukey post-test. (**F**) Densitometric quantification of the p-CrkII levels normalized to the GAPDH levels. The data corresponds to 3 independent experiments. Statistical analysis corresponds to ANOVA and Tukey post-test (0 min/15 min * *p* = 0.0197; 0 min/30 min. ** *p* = 0.003; 0 min/60 min * *p* = 0.0441).

**Figure 3 ijms-24-01944-f003:**
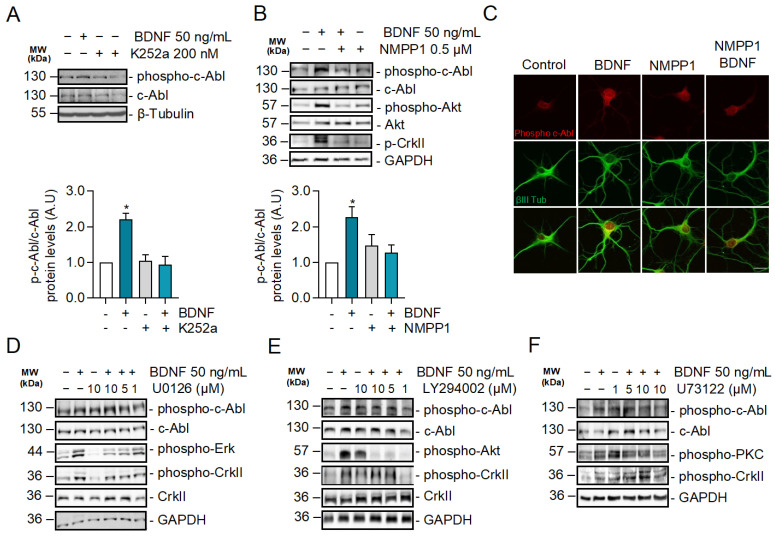
c-Abl activation induced by BDNF is dependent on the activity of TrkB receptors but does not require PI3K, ERK1/2, and PLC-γ-dependent signaling pathways. (**A**) Seven DIV neurons treated with BDNF in the presence or absence of K252a (200 nM). Phospho-c-Abl (Tyr-412), c-Abl, Phospho-CrkII (Tyr221), Phospho-Akt (Ser473), Akt, and GAPDH levels were analyzed by western blot. Densitometric quantification is shown below. The data corresponds to 3 independent experiments. Statistical analysis corresponds to ANOVA and Tukey post-test (**B**) Seven DIV neurons derived from TrkB^F616A^ mice were stimulated with BDNF in the presence or absence of 0.5 µM 1NM-PP1. Phospho-c-Abl (Tyr-412) and Phospho-CrkII (Tyr221), Phospho-Akt (Ser473), Akt, and GAPDH levels were evaluated by western blot. The data corresponds to 3 independent experiments. Statistical analysis corresponds to ANOVA and Tukey post-test (* *p* = 0.0213). (**C**) Seven DIV neurons from TrkB F616A mice were treated with BDNF in the presence or absence of 1NM-PP1. Representative images of immunofluorescence against Phospho-c-Abl (Tyr-412) in red and βIII tubulin in green. Scale bar = 10 µm. (**D**) Seven DIV rat primary hippocampal neurons treated with BDNF in the presence or absence of the MEK1/2 inhibitor U0126 (1, 5 and 10 µM). Phospho-c-Abl (Tyr-412), c-Abl, Phospho-CrkII (Tyr221), CrkII, Phospho-Erk1/2 (Thr202/Tyr204), and GAPDH levels were analyzed by western blot. *n* = 3. (**E**) Seven DIV rat primary hippocampal neurons were pretreated with 5 µM PI3K/Akt inhibitor, LY294002, for 60 min, and then stimulated with 50 ng/mL of BDNF for 30 min. Phospho-c-Abl (Tyr-412), c-Abl, Phospho-CrkII (Tyr221), CrkII, Phospho-Akt (Ser473), and GAPDH levels were analyzed by western blot. *n* = 3 (**F**) Seven DIV rat hippocampal neurons were pretreated with 5 µM PLC**-γ** inhibitor, U73122, for 60 min, and then stimulated with 50 ng/mL of BDNF for 30 min. Phospho-c-Abl (Tyr-412), c-Abl, Phospho-CrkII (Tyr221), Phospho-PKC (Thr-538), PKC, and GAPDH levels were analyzed by western blot. *n* = 2.

**Figure 4 ijms-24-01944-f004:**
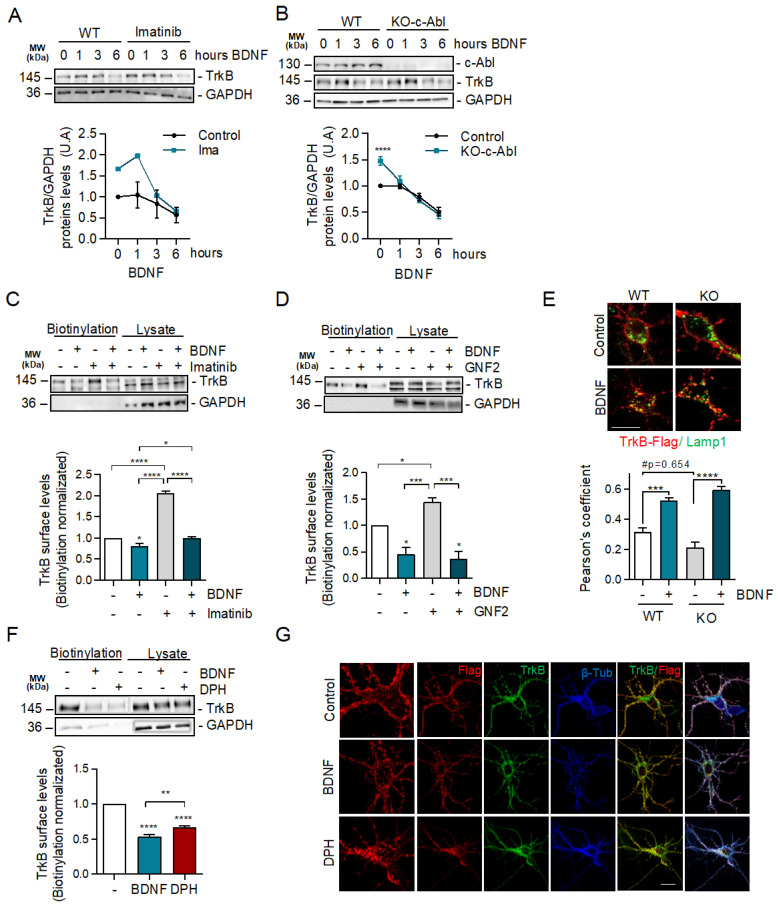
c-Abl activation downregulates basal surface TrkB levels but does not affect exogenous BDNF-induced TrkB endocytosis. (**A**) Seven DIV rat hippocampal neurons were treated with 5 µM Imatinib for 1 h, and then stimulated with 50 ng/mL of BDNF for 0, 1, 3, and 6 h. Protein extracts were immunoblotted for TrkB and GAPDH levels. The data corresponds to 4 independent experiments. (**B**) Seven DIV hippocampal neurons from c-Abl knockout mice (c-Ablloxp/c-Ablloxp; Nestin-Cre^+^) (c-Abl-KO) were treated with 50 ng/mL of BDNF for different amounts of time (1, 3, and 6 h). Lysates were analyzed by western blot for c-Abl, TrkB, and GAPDH. The data corresponds to 3 independent experiments. Statistical analysis corresponds to a two-way ANOVA, Sidak’s post-test (WT Control/KO Control **** *p* < 0.0001). (**C**,**D**) Rat primary hippocampal neurons (7 DIV) were pre-treated for 1 h with c-Abl inhibitors, and then treated with 50 ng/mL BDNF for one hour. Cell surface proteins were labeled by NHS-SS-biotin before initiation of TrkB internalization by BDNF at 37 °C for 30 min. Proteins on the surface were precipitated by streptavidin followed by western blot analysis using TrkB and GAPDH antibodies. The data corresponds to 3 independent experiments. (**C**) Representative immunoblot of cell surface TrkB receptor after 50 ng/mL BDNF and/or 5 µM Imatinib treatment. Densitometric quantification of TrkB levels showed an increase of TrkB levels on the plasma membrane in presence of Imatinib, but no differences were detected after BDNF stimulus. Statistical analysis corresponds to one-way ANOVA, Tukey’s post-test (Control/BDNF * *p* = 0.0317; Control/IMA and BDNF/IMA **** *p* < 0.0001; BDNF/IMA + BDNF * *p* = 0.0376). (**D**) Representative immunoblot of cell surface TrkB receptor after 50 ng/mL BDNF and/or 5 µM GNF2 treatment. Densitometric quantification showed a significant increase of TrkB levels on the plasma membrane in presence of GNF2, but no differences were detected after BDNF stimulus one-way ANOVA, Tukey’s post-test (Control/BDNF * *p* = 0.0263; Control/GNF2 * *p* = 0.117; BDNF/GNF2 *** *p* = 0.0007; GNF2/BDNF + GNF2 *** *p* = 0.0004). (**E**) Mouse primary hippocampal neurons (5 DIV) were transfected with TrkB-Flag. After 2 days of incubation, neurons were starved for 1 h. Then, neurons were washed and incubated with an anti-Flag antibody for 20 min at 4 °C. The cells were then treated with 50 ng/mL BDNF for 3 h. Subsequently, neurons were fixed and immunostained with anti-Lamp1. Quantification of colocalization between the TrkB signaling endosomes and lysosomes shows a decrease in control c-Abl KO neurons, but BDNF treatment abolished this difference. The data corresponds to 5 independent experiments. Two-way ANOVA, Turkey’s post-test (*** *p* = 0.004; **** *p* < 0.0001; # *p* = 0.0654). Scale bar = 10 μm (**F**) Representative immunoblot of cell surface TrkB receptor after 50 ng/mL BDNF or 5 µM DPH treatment. Densitometric quantification of TrkB levels in the plasma membrane showed that DPH treatment induces TrkB receptor internalization. The data corresponds to 3 independent experiments. Statistical analysis corresponds to one-way ANOVA, Tukey’s post-test (Control/BDNF and Control/DPH **** *p* < 0.0001; BDNF/DPH ** *p* = 0.0086). (**G**) Hippocampal neurons of 5 DIV were transfected with TrkB-Flag. After 2 days of transfection, neurons were incubated with anti-Flag antibodies (red) at 4 °C for 20 min. TrkB internalization was stimulated with 50 ng/mL BDNF or 5 µM DPH at 37 °C for 30 min and then the neurons were fixed, permeabilized, and immunostained with anti-TrkB (green) and β-tubulin (blue). Images were acquired on a confocal microscope. Scale bar = 10 μm.

**Figure 5 ijms-24-01944-f005:**
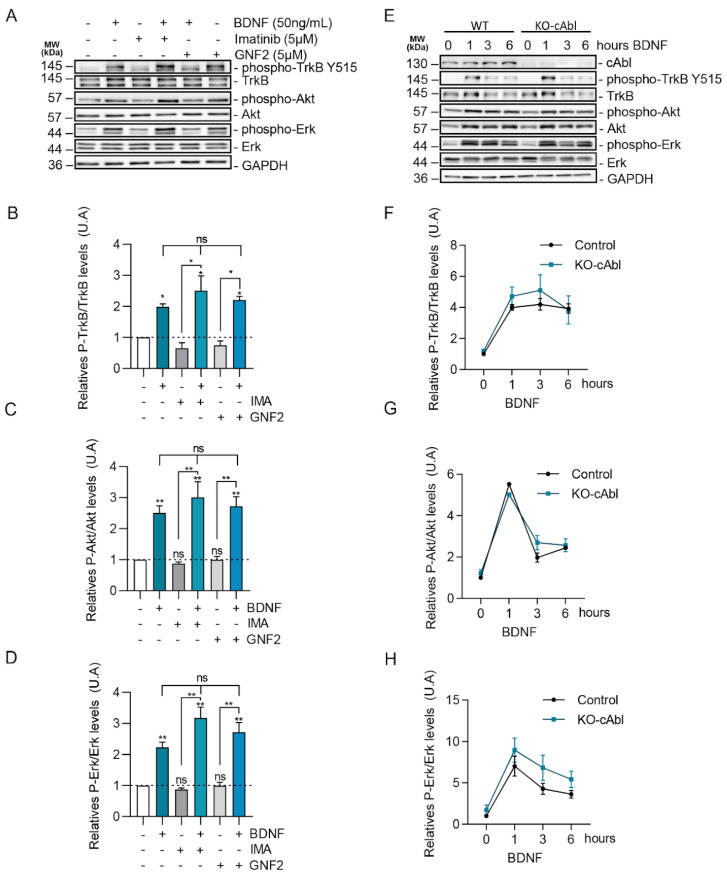
BDNF/TrkB downstream signaling activation is not affected by c-Abl activity and expression. (**A**) Seven DIV rat hippocampal neurons were pretreated with 5 µM Imatinib or GNF2, for 1 h, and then were stimulated with 50 ng/mL of BDNF for 1 h. Protein extracts were immunoblotted for phospho-TrkB (Tyr515), TrkB, phospho-Akt (Ser473), Akt, Phospho-ERK1/2 (Thr202/Tyr204), Erk1/2, and GAPDH levels. (**B**) Densitometric quantification of relative phospho-TrkB/TrkB levels. The data corresponds to 2 independent experiments. Statistical analysis corresponds to one-way ANOVA, Tukey’s post-test (Control/BDNF * *p* = 0.0098; Control/IMA + BDNF * *p* = 0.001; Control/GNF2 + BDNF * *p* = 0.0009; IMA/IMA + BDNF * *p* < 0.0001; GNF2/GNF2 + BDNF * *p* = 0.0001). (**C**) Densitometric quantification of relative phospho-Akt/Akt levels. The data corresponds to 3 independent experiments. Statistical analysis corresponds to one-way ANOVA, Tukey’s post-test (Control/BDNF ***p* = 0.0157; Control/IMA + BDNF ** *p* = 0.0017; Control/GNF2 + BDNF ** *p* = 0.006; IMA/IMA + BDNF ** *p* = 0.001; GNF2/GNF2 + BDNF ** *p* = 0.006). (**D**) Densitometric quantification of relative phospho-Erk/Erk levels. The data corresponds to 2 independent experiments. Statistical analysis corresponds to one-way ANOVA, Tukey’s post-test (Control/BDNF ** *p* = 0.0047; Control/IMA + BDNF ** *p* = 0.0015; Control/GNF2 + BDNF ** *p* = 0.0054; IMA/IMA + BDNF ** *p* = 0.0118; GNF2/GNF2 + BDNF ** *p* = 0.0175) (**E**) Seven DIV hippocampal neurons obtained from Abl knockout (c-Abl^loxp^/c-Abl^loxp^; Nestin-Cre^+^) (c-Abl-KO) were treated with 50 ng/mL of BDNF for different amounts of time (1, 3, and 6 h). Lysates were analyzed by western blot for c-Abl, Phospho-TrkB (Tyr-515), TrkB, Phospho-Akt (Ser-473), Akt, Phospho-Erk1/2 (Thr202/Tyr204), Erk1/2, and GAPDH. (**F**) Densitometric quantification of relative phospho-TrkB/TrkB levels. The data corresponds to 3 independent experiments. (**G**) Densitometric quantification of relative phospho-Akt/Akt levels. The data corresponds to 2 independent experiments. (**H**) Densitometric quantification of relative phospho-ERK/ERK levels. The data corresponds to 3 independent experiments. ns = Not statistically significant.

**Figure 6 ijms-24-01944-f006:**
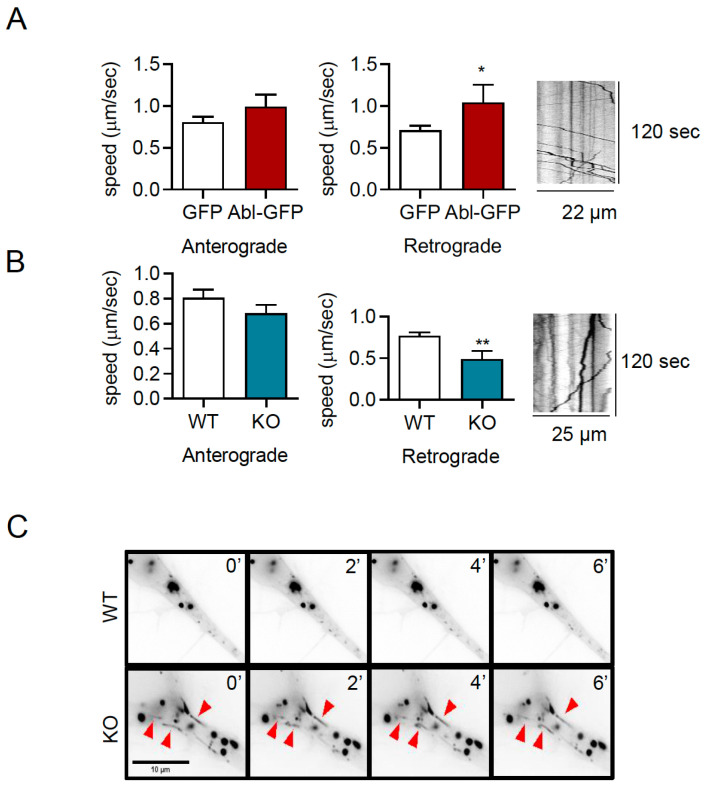
c-Abl increases the retrograde movement of TrkB-mCherry vesicles in dendrites. (**A**) Five DIV rat primary hippocampal neurons were transfected with TrkB-mCherry fusion protein and GFP or c-Abl-GFP fusion protein. After 2 days of incubation, the trafficking of TrkB-mCherry vesicles was evaluated by live cell imaging. Quantification of vesicular speeds showed that the overexpression of c-Abl-GFP increased the speed of retrograde movement. The data corresponds to 5 independent experiments. Statistical analysis corresponds to a Student’s *t*-test (* *p* = 0.047). A representative kymograph of this condition is presented. (**B**) Five DIV primary hippocampal neurons from c-Abl knockout and wild-type mouse were transfected with TrkB-mCherry, and after 2 days of incubation TrkB vesicular transport was evaluated by live-cell imaging. The retrograde speed of TrkB-m-cherry fusion protein is significantly reduced in c-Abl knockout neurons as compared to wild-type neurons. The data corresponds to 5 independent experiments. Statistical analysis corresponds to a Student’s *t*-test (** *p* = 0.0096) A representative kymograph of this condition is presented. (**C**) Representative image illustrating the shape of TrkB-mCherry fusion protein vesicles found in c-Abl knockout neurons, indicated by red arrows.

## Data Availability

The original contributions presented in the study are included in the article and Appendix A. Further inquiries can be directed to the corresponding authors.

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
