# Peer review of "c-Abl Tyrosine Kinase Is Required for BDNF-Induced Dendritic Branching and Growth"

_ijms, 2023, doi:10.3390/ijms24031944_

Round 1

Reviewer 1 Report

The work of Chandía-Cristi et al. establishes very convincingly that c-Abl is a downstream target of TrkB, not only by showing its BDNF-dependent phosphorylation and downstream activation, but also demonstrating that c-Abl is necessary and sufficient for BDNF-induced dendritic branching. The use of pharmacological inhibitors and genetically modified neurons build up a solid argument regarding the participation of c-Abl in shaping the complexity of the dendritic tree, although falls short to propose a mechanism that explain the role of c-Abl in the system. Indeed, despite showing changes in fast transport of dendritic endosomes containing internalised TrkB, c-Abl appears not necessary for TrkB endocytosis, seems to have no effect on the trafficking to late endosomes/lysosomes nor a clear place on the well-established signalling pathways. Crucially, it is not clear how the modulatory effect of c-Abl on dendritic transport of TrkB translates into its necessity for BDNF-induced dendritic branching —this being the most important gap of the work.   Major points:  
  1. The authors may well construct a narrative where the main claim of the paper becomes the fact that c-Abl constitutes a novel interactor and downstream signalling partner of TrkB, which appears to work independent of the best characterised signalling pathways. Then, use the requirement of c-Abl for BDNF-dependent dendritic arborisation as well as the sufficiency of c-Abl to promote branching on its own, as relevant examples of its role in neurotrophic signalling. However, should the authors want to focus on dendritic branching, they will need to provide additional evidence and significant discussion about potential mechanisms, given that most of the data is actually excluding potential players, rather than supporting a particular hypothesis.
  2. The authors briefly mentioned the possibility of the Cdk5 could be the downstream target of c-Abl that mediates this function by orchestrating cytoskeleton dynamics. This possibility seems particularly attractive, as it fits with the important role of cytoskeleton dynamics for dendritic branching, explains changes in endosomal transport and it may depend on local signalling changes that may not be detected by WB. Including an experiment to look at Cdk5 would strengthen the stance of this work, and in any case, discussing this hypothesis in detail seems absolutely essential.
  3. Independent of what the authors choose their narrative to be, the changes in endosomal transport of internalised TrkB in dendrites need to be discussed taking into consideration the available literature on signalling endosomes (i.e. role of Rab5 endosomes trafficking in dendrites) and their role on propagating neurotrophic signalling and activate nuclear effectors (i.e CREB activation has been shown to be required for BDNF-induced dendritic branching). Otherwise this very interesting finding looks rather disconnected of the main argument.
  4. Endocytosis and trafficking to LAMP1-positive endosomes may differ between soma and dendrites. In the light of the finding that the transport is altered, it would make sense to analyse those two compartments separately.
  5. Given that c-Abl appears to have a role on the localisation of TrkB to the plasma membrane on the steady state, but not affect the response to BDNF, it would be important to clarify whether this effect of c-Abl depends on endogenous BDNF (i.e. via an effect on BDNF secretion). Application of TrkB-Fc or blocking antibodies anti-BDNF may clear this point. Other explanations for the dissociation between surface localisation and no effect on endocytosis need to be discussed in the appropriate section.
  Statistics and visualisation:  
  1. No ANOVA test could have been correctly computed for the data on the plots of Fig 2 D and F; Fig 3 C, D and F; and Fig 4 C, D and F, given that there is no variance in the control group, breaking the homoscedasticity assumption of the ANOVA test. Please find an alternative way to normalise or analyse the dataset.
  2. Please justify the lack of quantification for the results in Fig3 D, E and F. Even if no statistics are done, a quantitative representation of the data (and the ratios!) would be very useful and increase readability.
  3. Please explain the line plot in Fig4E. Are these consecutive observations of the same population? Is there any other reason? Please justify the use of this particular method to measure co-localisation and include a detailed protocol in the methods section.
  4. Please report statistics with exact p values whenever possible.
  Minor points:  
  1. Lines 48-49 of the introduction include insufficient detail about the contribution of the cytoskeleton component to dendritic branching. Specially considering that this may end up being the most important piece of the proposed mechanism!
  2. It's not clear to me whether the labelling of the Figure 1LL is a cultural statement about spanish speakers (which I'd enthusiastically support if you also include Fig 1CH) or a typo that would need to be corrected for Fig 1M.
  3. Line 474-475 contains a duplicate
  4. Is "strikingly" the adverb you really want to use in line 475?

Author Response

The authors gratefully acknowledge the reviewers observations and comments, and will proceed to address them individually. 

Major points:

1. We restructured the discussion to explore possible mechanisms in more depth, and also to include evidence of Cdk5's possible contribution to c-Abl mediated dendritic branching (Lines 478-608). 

2. We performed western blot experiments to quantify Cdk5 and pCdk5 levels in neurons treated with BDNF and/or c-Abl inhibitors. Supplementary figure 5 shows a representative western blot and quantification of 2 independent experiments.  The results of the BDNF + GNF-2 - treated neurons support the possibility of c-Abl activating Cdk5 in response to BDNF treatment. Also, additional evidence and discussion were added to explore the role of Cdk5 (Lines 478-608)

3. A paragraph was dedicated to the discussion of the role of post-endocytic trafficking in BDNF-TrkB signaling (Lines 562-593). References 8 and 38 were discussed in additional detail to reinforce the importance of post-endocytic trafficking and signaling in the context of BDNF-TrkB signaling.

4. We agree with the observation that a separate analysis for dendritic and cell body endosomes could yield interesting results. However, the experiments were originally designed to analyze endosomal dynamics in neurites, and therefore we are unable to evaluate somatic endosomal dynamics at this point in time.

5. We were, unfortunately, unable to perform this experiment in the allotted time frame for the revision. 

Minor points: 

1. Lines 48-49 of the introduction include insufficient detail about the contribution of the cytoskeleton component to dendritic branching. Specially considering that this may end up being the most important piece of the proposed mechanism!

Lines 93-97 were added to explore the contribution of the cytoskeleton to dendritic branching. Additionally, references 15 and 17 were added as evidence to support this claim. 

2. It's not clear to me whether the labelling of the Figure 1LL is a cultural statement about spanish speakers (which I'd enthusiastically support if you also include Fig 1CH) or a typo that would need to be corrected for Fig 1M.

Corrected to 1M

3. Line 474-475 contains a duplicate

Eliminated duplicate

4. Is "strikingly" the adverb you really want to use in line 475?

Eliminated the adverb as internalization and phosphorylation are indeed 2 expected mechanisms of Trk receptor regulation.

  Statistics and visualisation: 

At this moment in time we have limited access to the raw data that contains the required information due to unforeseen circumstances, but expect to be able adequately respond to the reviewers concerns in a reasonable time frame. 

The line plot will be corrected to a more adequate graphical representation of the data, the exact p-values will be added where available and the statistics for Fig 2 D and F; Fig 3 C, D and F; and Fig 4 C, D and F will be revised and updated. 

Reviewer 2 Report

This manuscript by Chandía-Cristi et al. is very well written and the discussion is excellent. This is a very careful study and the data support the authors conclusions.  The authors convincingly show that BDNF-induced dendritic branching requires c-Abl expression and activity. The key and significant finding of this manuscript being that c-Abl promotes retrograde trafficking of TrkB.

Fig. 2B- should add BDNF label to lower panel of BDNF stimulated neurons.

Author Response

We gratefully acknowledge the reviewers comments and observations. 

  • The requested BDNF label for the lower panel of Fig. 2B, and also a Control label have been added. 

Reviewer 3 Report

Manuscript: c-Abl tyrosine kinase is required for BDNF-induced dendritic 2 branching and growth by América Chandía-Cristi et al. describes the search for new mechanisms related to TRkB signaling, combining the DNF/TrkB pathway with c-Abl in a context of dendritic branching and growth.  The research is very interesting and innovative, however, the description and presentation of several results need to be corrected.

Please describe in the methodology in how many repetitions blots were performed. Figures 2A and 3F require replacing the blots with better quality ones. Please include markers for all blots in the publication. All descriptions of the results calculated by one-way ANOVA or two-way ANOVA lack the statistical results presented as (F, n, p, SEM).

Author Response

We gratefully acknowledge the reviewers comments and observations on our work. We would like to address every comment individually:

  • We have included the number of western blot repetitions for each experiment in the figure descriptions, and 2 lines were added to the methods section to point the reader to the figure descriptions in order to find the number of independent experiments for any given experiment.
  • We are unable to provide other western blots to replace figures 2A and 3F, but believe that the images adequately display the results
  • Molecular weight markers and antibody markers were added to all the western blots presented in the paper.
  • Due to some unforeseen circumstances we currently have limited access to the raw data which contains the specific statistics results, but expect to be able to fix the format of the results as requested in a reasonable time frame.